# The Human Extracellular Matrix Diseasome Reveals Genotype–Phenotype Associations with Clinical Implications for Age-Related Diseases

**DOI:** 10.3390/biomedicines11041212

**Published:** 2023-04-19

**Authors:** Cyril Statzer, Karan Luthria, Arastu Sharma, Maricel G. Kann, Collin Y. Ewald

**Affiliations:** 1Department of Health Sciences and Technology, Institute of Translational Medicine, Eidgenössische Technische Hochschule Zürich, Schwerzenbach, CH-8603 Zurich, Switzerland; cyrilstatzer@gmx.ch (C.S.); arsharma@student.ethz.ch (A.S.); 2Department of Biological Sciences, University of Maryland, Baltimore County, 1000 Hilltop Circle, Baltimore, MD 21250, USA; karanl1@umbc.edu

**Keywords:** phenome, matrisome, matreotype, phenotype, extracellular matrix, data mining, SNP, PheWAS, GWAS, electronic health records, drug repurposing, precision medicine, collagen, human

## Abstract

The extracellular matrix (ECM) is earning an increasingly relevant role in many disease states and aging. The analysis of these disease states is possible with the GWAS and PheWAS methodologies, and through our analysis, we aimed to explore the relationships between polymorphisms in the compendium of ECM genes (i.e., matrisome genes) in various disease states. A significant contribution on the part of ECM polymorphisms is evident in various types of disease, particularly those in the core-matrisome genes. Our results confirm previous links to connective-tissue disorders but also unearth new and underexplored relationships with neurological, psychiatric, and age-related disease states. Through our analysis of the drug indications for gene–disease relationships, we identify numerous targets that may be repurposed for age-related pathologies. The identification of ECM polymorphisms and their contributions to disease will play an integral role in future therapeutic developments, drug repurposing, precision medicine, and personalized care.

## 1. Introduction

A major goal in biomedical research is to relate diseases or phenotypes to genotypes influenced by environmental factors. The linking of genetic variants to phenotypes, or vice versa, is the first step in the generation of hypotheses to determine the molecular drivers underlying diseases and expressed traits. Genome-wide association studies (GWAS) and phenome-wide association studies (PheWAS) have been instrumental in generating such hypotheses in an unbiased manner in order to identify the genetic basis of diseases or traits. Genome-wide association studies begin by comparing a large number of individuals with clinical manifestations or phenotypes to healthy individuals or individuals without these phenotypes. The whole-genome sequencing of these individuals allows the identification of genetic variants, usually single-nucleotide variants (SNVs). The association is then determined by the increased frequency of SNVs or the finding of a genetic variant in patients compared to healthy individuals. The use of GWAS is a “forward genetics” approach. By contrast, the use of PheWAS is a “reverse genetics” approach. Phenome-wide association studies start with a genetic variant and survey a large number of different phenotypes to identify one or multiple phenotypes that are statistically associated with this given genetic variant or SNV. Since association is not causation, the ultimate goal in this field is to find causally or mechanistically linked genotype–phenotype relationships. Additional experimental research with model organisms is required to further validate these mechanistic discoveries.

The recent emergence of medically relevant PheWAS was made possible by associating large human genetic information with electronic health records [1,2,3,4]. This required the dense phenotyping of patients and the collection of patient samples into biobank repositories to enable researchers to apply omics approaches, such as genomics, proteomics, and metabolomics [4]. The discovery of novel mechanisms is facilitated by integrating open-source data across species. For instance, the Monarch Initiative is a large collection of over two million genotypic–phenotypic associations that integrates data across hundreds of species from more than 30 different databases [5].

In this study, we mined the data from previous PheWAS–GWAS studies and other databases to establish the phenotypic landscape of extracellular matrix (ECM) genetic variants. The ECM has emerged as a novel target for healthy aging [6,7], and disturbances of ECM integrity can be major drivers of either health or disease [8,9]. With the aim of attenuating diseases, twenty-seven clinical trials are underway on eleven different molecular targets for modulating ECM stiffness as a primary outcome measure [10]. Given the clinical importance of the ECM, we hypothesized that variants of ECM genes are associated with disease phenotypes and other physiological traits. To address this, we used three key approaches. 

First, we surveyed published reports on genotype–phenotype interactions in variants of ECM genes associated with diseases. The group comprising all the possible gene products that either form extracellular matrices or are associated with or remodel ECMs is referred to as the matrisome [11]. The human matrisome consists of 1027 proteins [12]. The core matrisome, which includes the structural components of the ECM, consists of collagen, proteoglycan, and glycoproteins. The associated matrisome is mainly responsible for the maintenance, upkeep, and remodeling of the ECM and consists of ECM-related proteins, secreted factors, crosslinking proteins, and proteinases [12].

Second, we analyzed the variants in these matrisome genes using the clinical-variance-diseasome list to identify unique diseases with multiple SNVs in matrisome genes. Using the integrative disease data from the Monarch Initiative, we linked these diseases with specific genes in the matrisome. Using the PheWAS approach, we then analyzed the genes implicated in the phenotypic associations of disease states.

Third, we used this list of genes and diseases to create gene–disease associations based on disease categorization, from typical diseases to unexpected and age-related diseases. We identified a list of specific collagen genes that displayed the highest frequency of associations with multiple disease states per core-matrisome gene. Using these associations, we then analyzed the specific predominance of mutations in these collagen genes. By cross-referencing the GWAS-PheWAS database and the DrugBank repositories, we identified numerous drugs which may affect matrisome genes. Finally, we analyzed the rare-disease databases to identify associations with genes involving the matrisome.

Thus, we used the matrisome list as a basis for screening. Taken together, our computational efforts identified targets that can be validated by experimental approaches to gain novel mechanistic insights into matrix biology with translational value.

## 2. Materials and Methods

Data source is referenced in the result section. Data cleaning, analysis, and visualization were performed using the statistical programming language R, utilizing the key packages dplyr, purrr, and ggplot2, and in Python, utilizing the pandas, numbat, scipy, and dask packages. All processed and output data are provided in Appendix A.

To establish a robust disease–gene association, publicly available datasets were obtained using standard web download and combined as described in [13] (table updated Jan 2020, Appendix A). The associations include clinical-variance data on human diseases [14] and the causal human disease genes from the Monarch initiative (https://monarchinitiative.org/, diseases tab, human, causal genes) (accessed on 27 January 2020).

The diseases were subsequently categorized into three groups: ECM-associated illnesses (A), other common diseases (B), and age-related illnesses (C). The grouping was performed based on substring matching using the terms listed in Appendix A.

The human-matrisome-gene list was obtained from Naba et al. [12] and was utilized to subgroup the ECM-associated diseases by their matrisome division and category. The PheWAS of GWAS Catalog of SNVs was obtained from https://phewascatalog.org/ (accessed on 27 January 2020) (dataset “PheWAS of GWAS Catalog of SNPs”). Several of the graph-network visualizations were developed in Python utilizing the cytoscape and forceatlas packages.

The orphan disease data were accessed at Orphadata, and are available online: http://www.orphadata.org/cgi-bin/index.php (accessed on 16 February 2023).

## 3. Results

### 3.1. Clinical Implications of the Human Matrisome

Previously, more than 1000 mutations in collagens were implicated in about 20 common diseases [15,16]. Different types of collagens localize in varying types of tissue (Figure 1). To expand this, we first took advantage of a comparison of the 1027 human matrisome genes [12] with genome-wide association studies (GWAS) coupled to phenome-wide association studies (PheWas) of curated medical health records (resource: PheWas catalog [17]). Out of the 3144 single-nucleotide polymorphisms (SNVs) associated with clinical phenotypes [17], we uncovered 140 SNVs located within the core matrisome and matrisome-associated genes. The most notable findings were the striking associations of the *TNXB* gene, responsible for the glycoprotein tenascin XB, with celiac or tropical sprue, celiac disease, and, in particular, with diabetic type I neuropathy (*p* value < 5 × 10^−8^) (Figure 2, Appendix A). These matrisome proteins, containing disease-associated SNVs, function in large interactive complex ECM networks and may exhibit some interrelated pathologies through signaling-related characteristics. Extracellular-matrix proteins form protein–protein-interaction complexes. Mutations in either of these ECM proteins in a complex should manifest in a similar phenotype. Previously, the network of human proteins of genetic disorders was defined, i.e., the human phenome–interactome [18]. In total, 18 matrisome and adhesome proteins are predicted to be associated with the 506 protein–protein-interaction-based human disease complexes [18] (Appendix A). Interestingly, in addition to matrisome-gene products, two integrins (*ITGB7* and *ITGA2*) were found that might link the ECM to cellular signaling (Appendix A). To expand on this idea of using ECM complexes, we used the matrisome to screen previously predicted protein–protein interactions in the human disease network [14]. We found 305 unique matrisome genes involved in 542 matrisome–matrisome protein-interaction pairs with implications for human diseases (Appendix A). Out of the 3824 human-disease-network genes, we identified 161 matrisome genes associated with 270 human diseases (Appendix A), suggesting a significant contribution of these genes to human diseases.

### 3.2. Genetic Variants in Matrisome Genes Associated with Diseases

To gain a more global view of the human-disease landscape associated with SNVs in matrisome genes, we took three approaches. First, we used the clinical-variance-diseasome list (www.ncbi.nlm.nih.gov/clinvar/ (accessed on 27 January 2020) [13]). In this dataset, there are 1840 unique genes associated with 9281 polymorphisms, among which we found 181 unique matrisome genes associated with 1216 polymorphisms. Within this list of polymorphisms, we found that Alport syndrome displayed the largest occurrence, with 423 SNVs within the matrisome genes, followed by Marfan’s syndrome, with 149 SNVs, and osteogenesis imperfecta, as the third most commonly referenced disease, with 132 SNVs (Appendix A). 

Second, we used the integrative disease data from the Monarch Initiative, which accesses over 30 externally curated databases (https://monarchinitiative.org/disease (accessed on 30 January 2020) [5]). We found 285 matrisome genes associated with 553 human diseases (Appendix A). We identified a disease association in 30 out of the total 44 collagen genes in humans, resulting in 106 different pathologies (Appendix A). The collagens with the strongest known disease association are *COL2A1* and *COL7A1*, with 19 and 13 implicated collagenopathies, respectively. The majority of the disease associations, comprising 19.9% of all the collagen gene-to-disease links, were identified in the collagen *COL2A1* (Appendix A). This collagen is predominantly linked to different walking and growth disorders, resulting in the following conditions, among others: spondyloepimetaphyseal dysplasia (Strudwick type), which causes short stature and skeletal abnormalities; metaphyseal chondrodysplasia (Schmid Type), causing short limbs and short stature; spondyloepimetaphyseal dysplasia congenita, involving short stature and skeletal abnormalities; Kniest dysplasia, which causes skeletal, visual, and auditory abnormalities and short stature; hypochondrogenesis (achondrogenesis Type II), which produces short limbs and altered bone growth; platyspondylic dysplasia (Torrance type), with shortened-limb formation; spondyloperipherald dysplasia (short ulna syndrome); dysspondyloenchondromatosis; spondyloepiphyseal dysplasia (Stanescu type); familial avascular necrosis of the femoral head; and avascular necrosis of the femoral head (primary) [19].

Overlapping with its involvement in multiple skeletal phenotypes, this collagen is also involved in visual perception, leading to retinal thinning (multiple epiphyseal dysplasia, Beighton type), retinal tears (autosomal dominant rhegmatogenous retinal detachment), and retinal detachment (Stickler syndrome type 1; Appendix A). By contrast, Stickler syndrome has the most collagens (six) linked to this syndrome (*COL2A1, COL9A1, COL9A2, COL9A3, COL11A1,* and *COL11A2*; Appendix A), followed by Ullrich congenital muscular dystrophy, with four collagens associated (*COL6A1, COL6A2, COL6A3*, and *COL12A1*; Appendix A). 

Third, we used a PheWAS approach and searched the top 1000 hits with *p*-value < 10^−6^ from the UK Biobank, which consists of more than 2000 phenotypes associated with 3144 GWAS from more than 500,000 individuals [https://www.ukbiobank.ac.uk (accessed on 30 January 2020); [20]. We identified 61 matrisome genes in these top 1000 hits (Appendix A), including three collagens (Figure 3). Interestingly, *EYS* (eyes shut homolog) displayed a marked correlation with the primary cause of death from peripheral vascular disease. As previously noted in a similar linkage with the GWAS analysis of type I diabetic neuropathy, the use of insulin 1 year after diagnosis or the use of an insulin product is strongly linked with *TNFXB* variants. The disorders strongly linked with the three collagens, *COL11A2, COL13A1*, and *COL15A1*, also exhibit a strong correlation with other genes: hyperthyroidism thyrotoxicosis, with *MUC22*; malignant neoplasm of the thyroid gland, with *SLIT2*; malabsorption and coeliac disease, with *SFTA2*; and heart failure, with *GPC6* (Appendix A). 

Altogether, we surveyed multiple databases and public resources (clinical variance, human diseases, and Monarch diseases) for the association between matrisome genes and human diseases. Overall, we found 1142 human diseases associated with the matrisome (Appendix A). When considering the GWAS and sequencing data, we found that the human-disease matrisome consisted of 333 out of the total 1027 matrisome genes (32.4%) (Figure 4), suggesting the importance of the matrisome in human pathologies. The components of the core matrisome (collagens, glycoproteins, and proteoglycans) had the strongest links to specific diseases, suggesting the role of structural proteins in the human diseasome.

### 3.3. Matrisome in Age-Related Diseases

Having established the contribution of the variants of the matrisome genes to human diseases, we next aimed to establish which kinds of diseases are most heavily affected by the ECM. To this end, we manually curated the disease–gene associations from multiple data sources (see M&M, Appendix A). We curated 28,920 disease–gene associations from a total of 10,570 genes (Appendix A). Out of the 1027 human-matrisome genes, 656 corresponded to 2825 matrisome-gene–disease associations, which represented about 10% of all our curated disease–gene associations (Appendix A). Reading through these 2825 matrisome disease–gene associations, we noticed the repeated emergence of three kinds of disease, and we subjectively grouped them into three main disease categories (A–C; see Appendix A for details on selected disease terms).

Category (A) comprised expected and typical ECM diseases, such as Ehlers–Danlos syndrome, osteogenesis imperfecta, Marfan syndrome, Alport syndrome, Fraser syndrome, Von Willebrand disease, etc. (Category A = connective-tissue diseases).

In category (B), we grouped common diseases in which alterations in the ECM have not been directly implicated but could potentially be affected by faulty ECMs, such as diabetes type 1, asthma, autism, lissencephaly, schizophrenia, seizures, muscular dystrophy, obesity, stroke, etc. (Category B = common diseases).

Since we noticed several chronic and age-related pathologies in the 2825 matrisome-gene–disease associations, we grouped the age-related diseases in category (C), including arthritis, Alzheimer’s disease, cancer, diabetes type 2, chronic obstructive pulmonary disease, fibrosis, Parkinson’s disease, cirrhosis, osteoporosis, hypertension, etc. (Category C = age-related diseases).

Surprisingly, age-related diseases formed the predominant category for the whole matrisome, as well as for the core matrisome (Figure 5, Appendix A, Appendix A). Out of the 2825 matrisome disease–gene associations, we found 1250 age-related diseases (category C), including overlapping disease categories (A_C, B_C, and A_B_C; Figure 5, Appendix A). Among the age-related diseases, about 15–30 matrisome genes were each associated with different cancers, such as prostate cancer, lung cancer, colon cancer, and glaucoma (Figure 6A, Appendix A). A number of matrisome genes (5–20) were each associated with age-related diseases, such as macular dystrophy, cardiovascular diseases, metabolic diseases (diabetes type II and severe obesity), Alzheimer’s disease, and fibrotic lung diseases (Figure 6A, Appendix A). Most surprisingly, about 150 matrisome genes were associated with autism, about 50 were associated with schizophrenia, and 35 were associated with intellectual disabilities (Figure 6A, Appendix A). The sheer frequency of the involvement of matrisome genes in type B (unexpected) and type C (age-related) diseases strongly implicates ECM proteins in disease association, suggesting that each individual gene may be associated with a plethora of different disease states.

Next, we determined the matrisome genes associated with several different diseases. Basement-membrane-forming collagen type IV (*COL4A1*) and laminin (*LAMA2*), which form ECMS around organs [21], are associated with 15 and 13 different diseases, respectively (Figure 6B, Appendix A). They are followed by fibril-forming collagen type III (*COL3A1*), collagen type I (*COL1A1, COL1A2*), collagen type V (*COL5A1, COL5A2*), and collagen type II (*COL2A1*), which form the connective tissues that support the muscles, joints, skin, and other organs [21] (Figure 6B, Appendix A). Fibrillin (*FBN1, FBN2*), which are components of the elastic fibers in the cardiovascular system, are implicated in more than eight diseases (Figure 6B, Appendix A), but fibrillins also attach to the large latent protein complex (*LTBP2*) and TGF-1β (*TGF1B*) (Figure 6B, Appendix A), which is important for collagen production and ECM remodeling [22]. Thus, we found a strong association between collagen-forming ECM and several diseases.

### 3.4. Collagens and Diseases

To gain a better understanding of the relationship between collagens and diseases, we performed a cluster analysis. We found an expected correlation of *COL4* with Alport syndrome, and an unexpected correlation between hearing loss and chronic kidney disease. A lower waist-to-hip ratio was associated with *COL6*, as was expected due to the associations with adipose-tissue fibrosis and metabolic dysregulation [23], and it was a predictor of all-cause mortality in type 2 diabetes and microalbuminuria [24]. Surprisingly, *COL12* was linked with early-onset PD, and *COL18* was associated with progressive neurodegenerative disease. In addition, osteoarthritis and schizophrenia were linked to variants of multiple types of collagens (Figure 7, Appendix A).

Next, we established which amino acid in collagens is the most frequently mutated and associated with diseases. Glycine was by far the most heavily disease-associated mutation, followed by proline and arginine (Figure 8A, Appendix A). For steric reasons, glycine is the smallest amino acid and is required to be in the third position of the (Gly-X-Y) repeats, where X is often proline, and Y is often hydroxyproline [21]. Proline in the endoplasmic reticulum is post-translationally modified to hydroxyproline, which is important for stabilizing the collagen triple helix [21]. Interestingly, the most frequent substitution of an initial glycine is by arginine, followed by aspartic acid, serine, and valine (Figure 8B, Appendix A).

### 3.5. Potential Strategies Using Matrisome for Drug Repurposing

Given the large contribution of the matrisome to diseases, we examined whether the matrisome can provide targets for drug-repurposing strategies. The key to drug repositioning is to cross-reference GWAS-PheWAS associated with diseases with DrugBank repositories [25]. With this approach, about 15,000 drug–disease relationships and over 38,000 novel drug-repurposing candidates were identified in silico [25]. On this basis, we found five matrisome genes (*COL1A2*, *NOV*, *LPA*, *MMP24*, and *PLG*) with 13 distinct drug candidates for repurposing (Appendix A). Strikingly, nine of these drugs can be used to target plasminogen (*PLG*) rs783147 SNV in at least sixteen diseases (Appendix A; see Discussion for further details). Thus, there is untapped potential for the repurposing of drugs targeting the matrisome.

### 3.6. Targeting Matrisome Proteins in Rare Diseases

Unlike cancer, cardiovascular diseases, diabetes, and other highly prevalent diseases, rare and orphan diseases still lack investment in medical research, drug development, and specialist knowledge. Around 7000 diseases are classified as “Rare and Orphan diseases,” and their number is increasing by almost 300 every year [26] (www.rarediseases.org) (accessed on 29 January 2020). A disease qualifies as rare if fewer than 1 in 2000 people are affected [27]. Approximately 80% of rare and orphan diseases are caused by genetic mutations [28], and we sought to determine how many of these mutations are in the matrisome. We found 311 matrisome genes linked to 460 unique rare diseases (Appendix A).

## 4. Discussion

While the use of GWAS and PheWAS is an integral tool for genetic epidemiological research and the elucidation of pathways that may contribute to disease pathology, further applications can be pursued. In the context of matrisomal genealogy, we aimed to unearth the implications of ECM matrisome genotypes and SNV variants for human diseases, ultimately producing the quantified ECM diseasome. The capabilities of GWAS and PheWAS have also enabled the linking of diseases as co-morbidities [29]. Furthermore, GWAS and PheWAS techniques also uncover a large network of potential drug targets that may be repurposed to target specific diseases that display strong redundant associations with certain polymorphisms, diversifying the array of pharmacological therapeutic targets available for exploration [30].

Among the total of 1027 matrisome genes that were analyzed, 140 SNVs were found. The most significant association was that of diabetic type I neuropathy with the TNXB gene, which is responsible for the production of the glycoprotein tenascin XB. The TNXB gene was previously shown to be moderately implicated in type I diabetes, but a further analysis of this association is warranted regarding the HLA region and non-HLA class II genes [31]. Elevated serum tenascin-C (*TNC*) levels were shown to be indicators of increased risk of cardiovascular events and death from type II diabetes, but the role of TNX in these processes must be further elucidated [32]. The *TNXB* gene was linked to Ehlers–Danlos syndrome due to the lack of organizational structure in the collagen framework within the ECM, and the association with diabetic type I neuropathy may be due to peripheral axonal stretching and pressure, increasing susceptibility to neuropathic pathology [33]. 

The ECM interactome is a target of growing interest in the context of numerous diseases. The ECM-signaling interactions regarding movement, adhesion, and growth are implicated in breast cancer, along with collagens and fibrinogen [34]. Cell-matrix and matrisome protein–protein interactions are evidently associated with numerous diseases, as many of the diseases analyzed (270 out of 3824) were linked to 161 matrisome genes. The prevalence of the ECM interactome within disease networks must be explored further, as ECM-cell, -matrix, and -signaling interactions were previously described as drivers of mammalian disease [35,36]. 

The linkage of various matrisome SNVs with Alport syndrome [37], Marfan syndrome [38], and osteogenesis imperfecta [39] is consistent with previous association studies, as these diseases are indicative of collagen irregularities or connective-tissue dysfunction. Interestingly, the majority of collagen-gene types (30 out of 44) are associated with disease, particularly *COL2A1* and *COL7A1*. The *COL2A1* collagen is linked to a large variation in phenotypic displays, ranging from growth and short-stature disorders to retinal dysfunction. The *TNXB* gene was additionally linked, through PheWAS, with the use of insulin 1 year after the diagnosis of diabetes and the general use of insulin, further implicating *TNXB* in the pathology of diabetes and diabetic neuropathy, as previously shown. The importance of the core matrisome in disease pathology is apparent, as out of the 333 genes in the disease matrisome, the majority are structural in nature. These structural components may be crucial in the molecular pathologies of the associated diseases. 

To further explore these links, the division of these genes into the aforementioned categories (A, B, C, and their combinations) provided insights into how many diseases link to matrisome genes. Although age-related diseases formed the majority of the associations with the matrisome genes, the most surprising result was the overwhelming number of associations with psychiatric and neurological disorders, autism/autism-spectrum disorders, schizophrenia, and intellectual disability. The role of the ECM in neurodevelopmental disorders has not been thoroughly stipulated and requires the use of iPSC-based methods, and prior research is significantly lacking. Certain manipulations of ECM components in vitro have resulted in alterations in the mechanical properties of the neocortex, but specific factors and the results of their modulation in the context of disease are still not apparent [40]. 

Interestingly, certain non-neurological components of ECM physiological phenotypes are apparent in individuals with ASD; children with ASD exhibit altered platelet functionality rather than platelet morphology compared to undiagnosed individuals due to increased collagen–ADP and collagen–epinephrine closure times, suggesting the involvement of matrisomal alterations in the disease [41]. Ehlers–Danlos syndrome, one of the expected matrisome-related diseases and hits reconfirmed in our analysis of gene–disease networks, also shares vast phenotypic overlap with ASD and generalized hypermobility spectrum disorders (gHSDs), presenting in the form of comorbidities and co-occurrences within families of diagnosed individuals, along with other comorbidities, such as intellectual learning disorders or ADHD [42]. Autism-spectrum disorder and comorbid ADHD are significantly associated with individuals who also have generalized joint hypermobility (GJH), and the diagnosis of GJH may even serve as a biomarker for future diagnoses of ASD and comorbid ADHD [43]. The additional role of core-matrisome-gene perturbations in proteoglycans may be another indicator of other disorders due to the high level of involvement of glycans in neurodevelopmental processes, warranting further exploration of the glycosylation of the ECM in neurodevelopmental disease [44]. 

Schizophrenia exhibited the second-highest number of gene associations after ASD. The ECMs of individuals afflicted with schizophrenia exhibit altered GABAergic signaling, chondroitin sulfate proteoglycans, MMPs, and maintenance, resulting in neuronal abnormalities [45,46]. Matrisome-gene expression is also disrupted in the ECM cortical areas of individuals afflicted with schizophrenia [47]. Neuronal migration and glial abnormalities due to dysfunctional reelin and chondroitin-sulfate proteoglycans may be additional contributors to the pathology of schizophrenia [48]. Given the role of the ECM in schizophrenic pathology, numerous matrisome-related targets may be explored for the treatment of this disease [49]. 

These ECM dysfunctions are not only specific to neurodevelopmental disorders; ECM components are highly dysregulated even in age-related neurodegenerative disorders, such as Alzheimer’s disease and Parkinson’s disease. The primary pathological manifestation of AD, the aggregates of amyloid beta (aβ) and the amyloid precursor protein (*APP*), are influenced by dysregulated chondroitin sulfates and heparan sulfates, which are associated with a higher protein-aggregate burden, along with increased amounts of the matrisome components tenascin, integrin, laminin, and galectin [50]. In PD, increases in the expression of collagen type I are evident, along with other pro-inflammatory changes in the surrounding matrix [51]. These changes may be attributed to the SNVs in the matrisome genes associated with AD and PD, increasing the susceptibility of individuals to the development of neurodegenerative disorders with increasing age. Exploring the matrisome gene and neurodegenerative-disease relationships within the context of dysregulated ECM mechanics may provide further targets for the development of pharmacological therapies.

Our cluster analysis of diseases with collagen variants aptly explains the involvement of core-matrisome-collagen genes in numerous diseases. Along with the previously expected associations with *COL4* and *COL6*, a few examples which may help to elucidate additional disease pathologies are apparent, such as the associations of *COL12* and *COL18* with neurodegenerative disorders. The *COL6* collagen is also important for Schwann-cell differentiation in the peripheral nervous system, but the role of *COL6* in the central nervous system must be further explored [52]. The *COL12* collagen, which is associated with early onset PD, was implicated in myopathy, but its role in age-related neurodegenerative disorders has not been elucidated [53]. Age-related changes in the ECM, such as collagen degradation, elastase upregulation, and fibronectin upregulation, may prime the cellular environment for increased risk of disease development and accelerated aging pathology, and the crossover of certain attributes of diseases such as EDS and Marfan syndrome with the ECM aging phenotype may provide insights into how aged ECMs and certain disease states may communicate [54]. Specific genetic associations of collagens may indicate an increased incidence of healthier aging, in which *COL1A1* rs107946 may suggest accelerated osteoporotic-related aging, but *COL1A2* rs3917 suggests a reduced risk of osteoporosis [55]. The interwoven diseases in Figure 7 and other related associations may be crucial to exploring and expounding ECM-related changes in healthy aging, particularly regarding how certain genetic predispositions may alter the quality-of-life course. 

When analyzing the most frequently substituted amino acid in the collagen chain, we found that glycine was most often substituted by arginine. The overwhelming display of this specific substitution cohesively strengthens the role of the understanding of genetic variants within the matrisome, particularly in collagen-related disorders. The substitution of glycine 661 with arginine in the *COL3A1* helix results in increased intracellular retention of the protein with abnormal thermal stability, resulting in EDS [56]. In Alport syndrome, the substitution of glycine 852 and 325 by arginine is evident in the *COL4A5* gene, which is responsible for basement membrane formation [57,58]. The glycine–arginine substitution is also evident in dominant dystrophic epidermolysis bullosa, with the variant of *COL7A1* [59]. This specific substitution has also been implicated in OI [60], particularly in the *COL1A1* and *COL1A2* types [61]. As these results are consistent with the involvement of specific amino-acid substitutions in typical ECM-expected disease, further analyses of the unexpected and age-related genetic variants are warranted to uncover how specific amino-acid substitutions may alter aging phenotypes. 

The extension of the application of GWAS and PheWAS techniques to matrisome genes allows the identification and repurposing of drugs to target specific polymorphisms, which may then ameliorate disease pathology. One of our significant target hits, the plasminogen (*PLG*) SNV rs783147, can be targeted for 16 disease indications. Plasminogen is a pro-fibrinolytic factor that is crucial for the removal of blood clots through its cleavage to the active enzyme plasmin [62]. Nine drugs, alteplase, aminocaproic acid, anistreplase, aprotinin, reteplase, streptokinase, tenecteplase, tranexamic acid, and urokinase, can be used to target certain complications, which may be particularly age-related, while many of these drugs are indicated for blood clotting and myocardial infarction or other cardiovascular complications, which are some of the leading causes of death [63]. Aging is a substantial risk factor for thrombosis and other thrombotic-related events, and with the addition of heightened interleukin 6 (IL-6) and C-reactive protein (CRP) levels, the risk of cardiovascular events is exacerbated in elderly individuals [64]. Plasminogen activator inhibitor 1 (PAI-1), a fibrinolysis inhibitor, is implicated in many age-related metabolic disorders and cancer [65], and PAI-1 levels significantly increase with age, predisposing elderly individuals to cardiovascular complications resulting from thrombosis and atherosclerosis [66]. Plasminogen and the inhibition of fibrinolytic activity may be opportune targets for ameliorating cardiovascular aging through the repurposing of identified drugs [67]. Analyses of drug repurposing can help to influence the trajectory of high-throughput drug screens for ECM-related pathologies [68], and can even link other increasingly age-related diseases to matrisome defects, such as cardiovascular disease [69]. 

In addition to the age-related implications of the matrisome and disease networks discussed above, the majority of the genes in the matrisome diseasome are linked to 460 rare diseases. Our findings strongly link polymorphisms located in the matrisome with many connective-tissue disorders, age-related diseases, rare diseases, and neurodevelopmental disorders. The emergence of omics technologies in combination with association studies may serve as an essential component in the further implication of the ECM in diseases. As shown in squamous cell carcinoma (SqCC), the integration of ECM features in aging and diseased tissue with multi-omics data on SqCC can aptly determine the risk of cancer development [70]. The applications for precision medicine are endless, as organoid models created with patient-derived cell lines and their associated SNVs and other manipulations of ECMs, which may be representative of certain phenotypes, can be further explored in disease pathology, particularly in other age-related diseases, inflammatory conditions [71], osteoarthritis [72], and processes such as tumor–stroma interactions [73] and various types of cancer [74]. The matrisome is earning an increasingly prominent role in the pathology of disease (Figure 9). The emergence of strong associations between collagenous dysfunction and related SNVs in disease is steadily increasing, and the associations of genetic predispositions to age-related phenotypes with variations in matrisomal composition may play a pivotal role in the discovery of therapies and treatments for the process of aging.

## Figures and Tables

**Figure 1 biomedicines-11-01212-f001:**
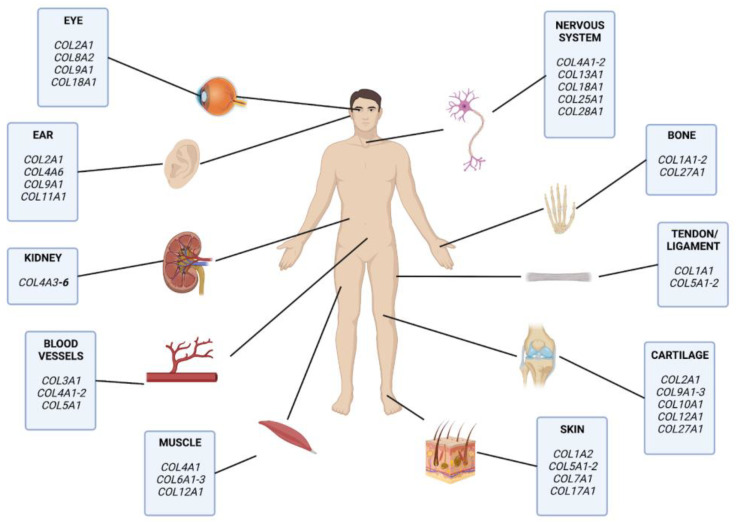
Collagens in Human Tissues. Different types of collagen proteins experience localized expression in various tissue types. For example, *COL2A1* is expressed in tissues with substantial components of cartilaginous fibers.

**Figure 2 biomedicines-11-01212-f002:**
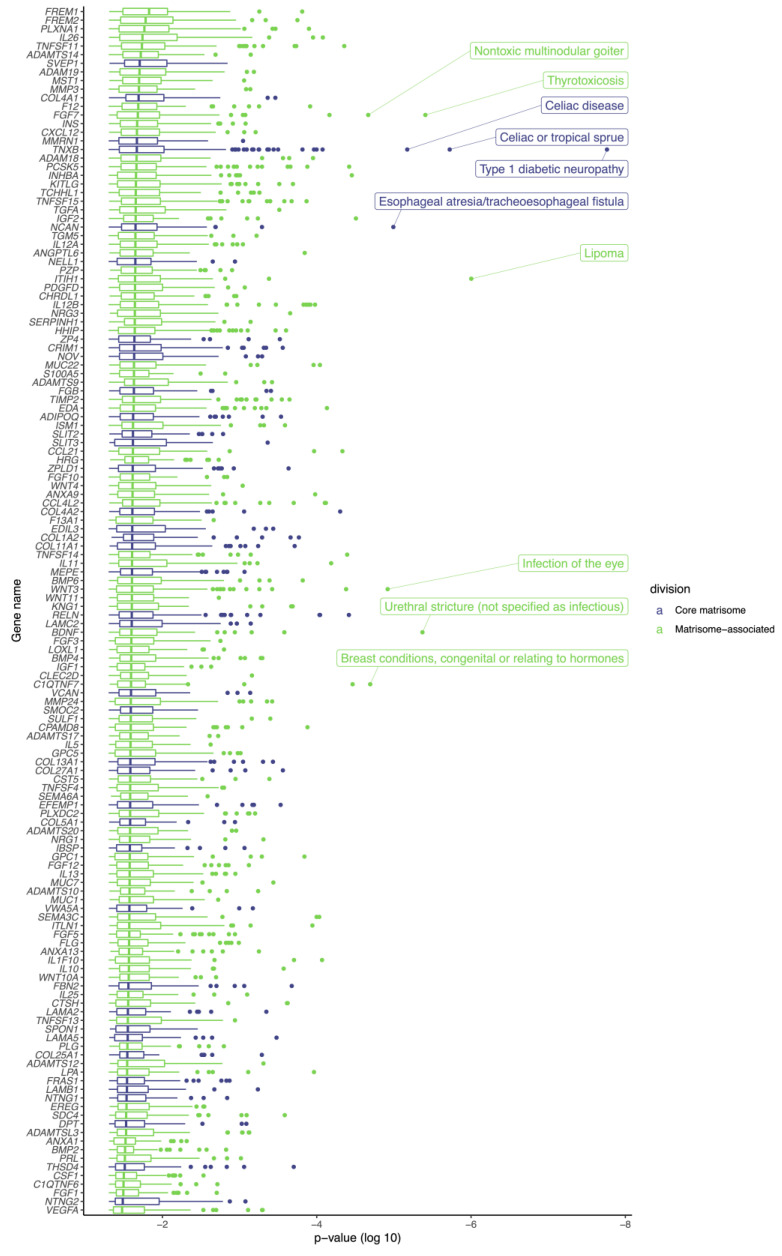
PheWAS of GWAS Catalog of SNVs. The human-matrisome genes are displayed on the y-axis, and the log10 *p*-value distribution of the associated phenotypes are shown as boxplots on the x-axis. Outliers are displayed as points, and only significantly associated phenotypes (*p*-values < 0.05) are shown. The ten most significantly associated phenotypes (*p*-value < 0.00001) are labeled. For more details, please see Appendix A. The phenotypes are colored according to the matrisome division with which the respective gene is associated (core matrisome genes in blue, matrisome-associated genes in green).

**Figure 3 biomedicines-11-01212-f003:**
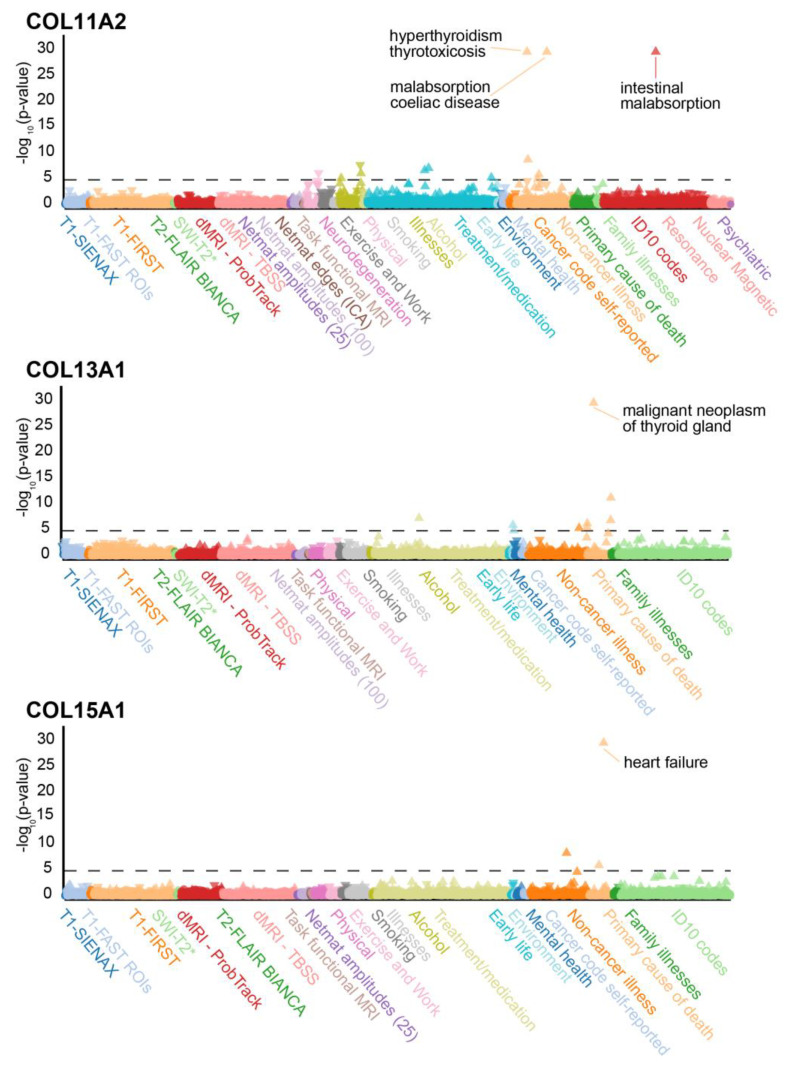
PheWAS analysis of *COL11A1, COL13A1*, and *COL15A1*. Individual phenotypes are on the x-axis, and the -log10 *p*-value is on the y-axis. The most significantly enriched phenotypes (*p*-value < 10^−20^) are labeled for each collagen. The dashed line represents the significance cut-off at a *p*-value of 10^−5^. Phenotypes are colored according to their phenotype category (Appendix A).

**Figure 4 biomedicines-11-01212-f004:**
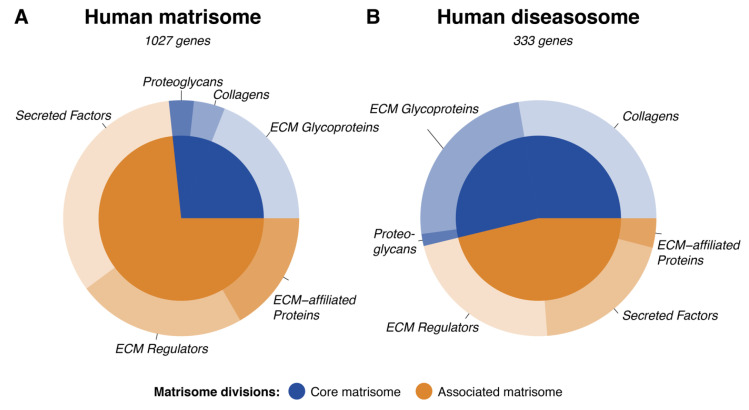
Involvement of the Human Matrisome in Pathology. The composition of the human matrisome is illustrated using a circular diagram reflecting the relative abundances of the core (blue) and associated (brown) matrisome divisions in the overall matrisome (**A**) and in the part of the matrisome which is specific to disease ((**B**), Appendix A). The divisions are further subdivided by the matrisome categories they contain. The subset of the human matrisome (1027 genes) responsible for the diseases (333 genes) is approximately 30% of the total and is enriched with collagens. This overrepresentation of collagens further highlights the importance of collagenopathies.

**Figure 5 biomedicines-11-01212-f005:**
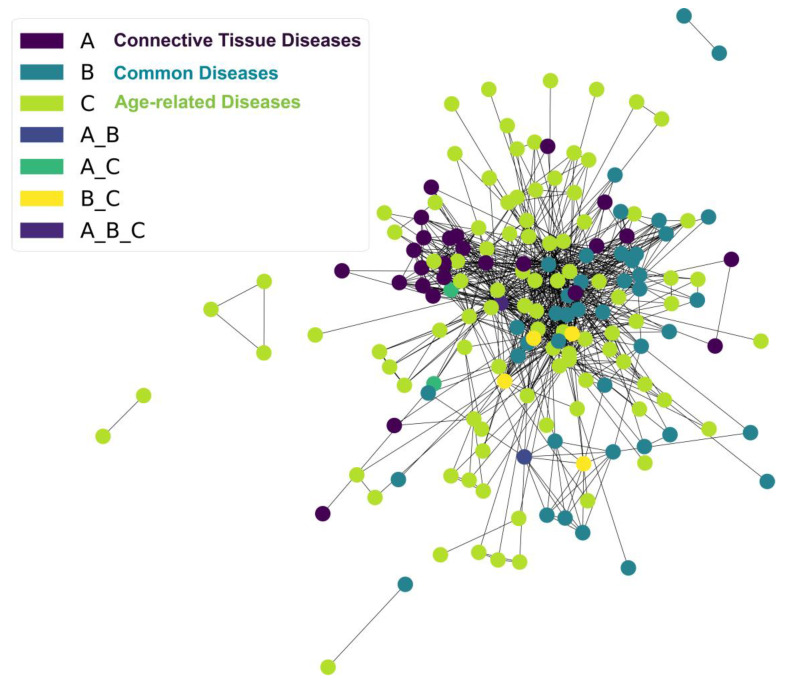
Total Matrisome Disease–Gene Associations. Cluster analysis of matrisome disease-gene–associations, composed primarily of age-related diseases and overlapping categories. For further details, including a graphical representation with the labeling of diseases and a tabular view, please see Appendix A.

**Figure 6 biomedicines-11-01212-f006:**
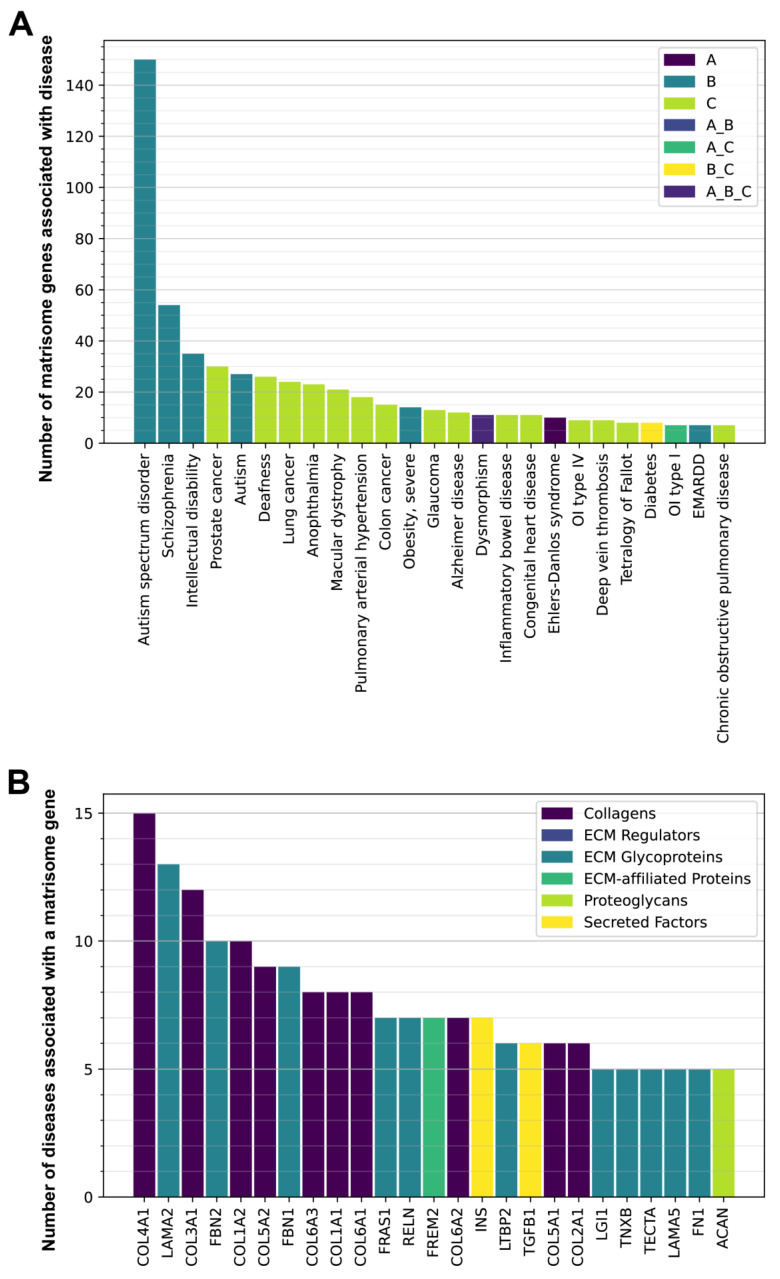
Matrisome-Gene–Disease Associations. Numerous diseases on the x-axis display an association with a certain number of matrisome genes on the y-axis (**A**). For example, autism-spectrum disorder, schizophrenia, and intellectual disability display the largest number of associated matrisome genes and belong to group B of diseases that would not be especially expected to correlate strongly with matrisome-related SNVs. The various matrisome genes on the x-axis (**B**) are associated with more than one displayed disease state (Appendix A).

**Figure 7 biomedicines-11-01212-f007:**
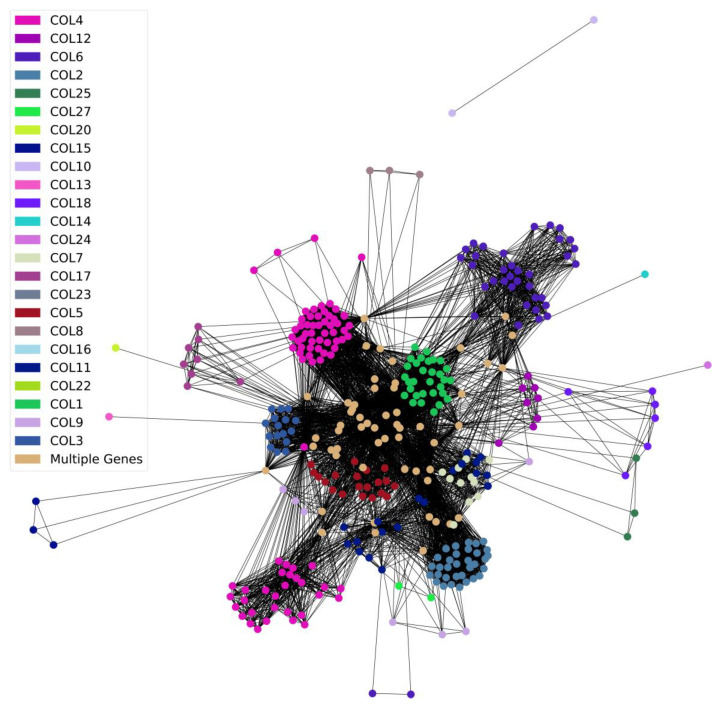
Disease–Collagen–Gene Associations. Numerous disease states are associated with various mutations in different collagen genes. Groups of diseases within the collagen diseasome formed a network in this cluster analysis due to their relatedness to a mutation in specific collagens. Multiple diseases, shown in beige, share numerous mutations in different types of collagen genes, which surround the beige centroid (Appendix A). For further details, including a graphic representation with the labeling of diseases and a tabular view, please see Appendix A.

**Figure 8 biomedicines-11-01212-f008:**
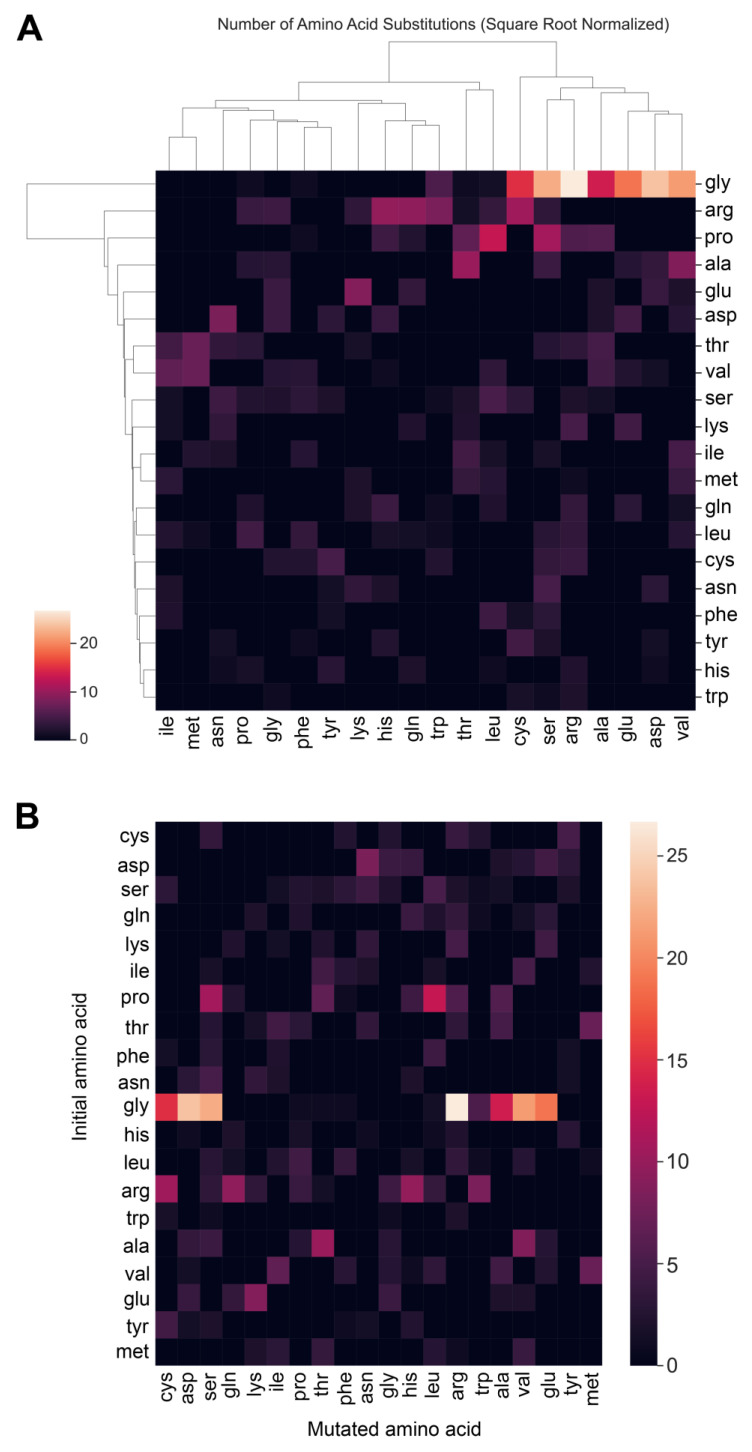
Predominant Mutations in the Collagen Helix. Heatmaps display the (**A**) frequency of amino-acid substitution, where glycine is the most substituted amino acid, and (**B**) the most frequent mutation substitution of glycine for arginine. Highest values indicate predominant amino-acid mutation (Appendix A).

**Figure 9 biomedicines-11-01212-f009:**
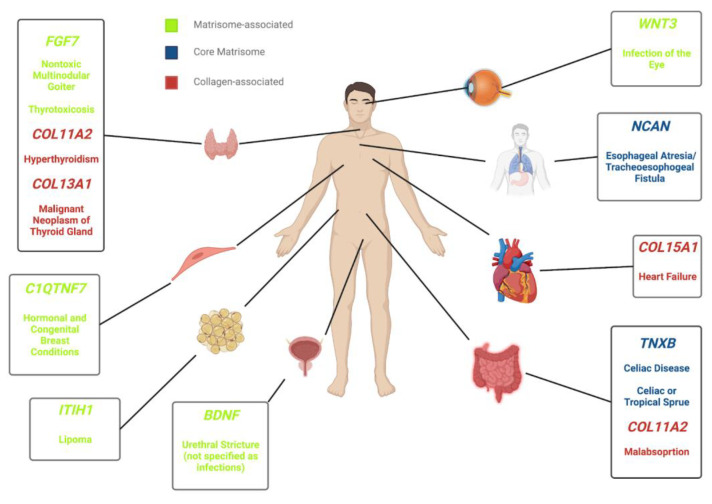
The ECM-Diseasome Body Map. An example of a proposed body map of matrisome genes that are associated with numerous disease states. Certain matrisome genes localize to various tissues in the body and may be linked to more than one disease state, suggesting therapeutic repurposing. This body map can be expanded, and various other body maps can be created through other gene–disease associations, which may be explored in the future.

## Data Availability

All data are available in Appendix A.

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
