# Peer review of "The Human Extracellular Matrix Diseasome Reveals Genotype–Phenotype Associations with Clinical Implications for Age-Related Diseases"

_biomedicines, 2023, doi:10.3390/biomedicines11041212_

Round 1

Reviewer 1 Report

 Summary

The authors are interested in the relationship between Extracellular Matrix (ECM) gene polymorphisms and disease states relevant to ageing. In particular the authors sought to utilise Genome-wide association studies (GWAS) and phenome-wide association studies (PheWAS) methodologies to perform this type of analysis. The authors use data from previous PheWAS-GWAS studies and other databases and focus on the matrisome i.e. the compliment of ECM proteins or genes expressed in a particular setting. Using a variety of approaches the authors create a set of matrisome genes linked to disease. Interestingly they take this data one step further via cross refence to DrugBank repositories and identified putative drugs that may target matrisome components.

The corresponding author is an expert in ECM analysis in the context of ageing, from a developmental perspective using the model organism C. Elegans, and the lead author has also published in this field previously (Matrix Biology Plus 2020). As such they are well placed to perform such a study.

The analysis provided in the manuscript is performed to a high standard and should be of interest to the ECM field and pharmaceutical industry. This reviewer is supportive of publication and has highlighted some issues, listed below, that could be used to potentially improve readability and data presentation.

Issues:

1.     1. Figure 1 outlines collagen expression in human tissues but the 1st in text reference to figure 1 (line 109-110) indicates that the figure displays mutations in collagen linked to common diseases. Please either provide a figure to illustrate this or adjust the text to reflect the content of the figure.

2.     2. Figure 2. PheWAS of GWAS Catalog of SNVs. Does this figure convey the information needed in the most readable format? It is difficult to read across from the highlighted disease associations to the genes. Moreover, is it useful to display the data for all genes without the disease association? This is provided in the supplementary table but it’s not clear how the 10 most significant terms have been selected. It seems to relate to the p-values (column M) in Table S1 but this is not stated. Perhaps you could provide a reduced figure 1 with additional PheWAS association categories? You should think about what is the most important information that you want to convey (genes or associations) and make sure that is provided. The rest of the data can be found in Table S1

3.     Figure 4 and at line 204 the authors state “The human disease-matrisome consists of 333 out of the total 1027 matrisome genes (32.4%) (Fig. 4), implicating the importance of matrisome in human pathologies.” Whilst this analysis does support the statement that ECM genes are important for human disease, does it underplay it’s relevance, as the analysis relies on the association of SNVs whereas disease association of ECM genes/proteins may additionally arise from over or under expression, aberrant localisation or copy number variation. Could the language around this discussion be altered to convey the complexity of this situation?

4.     Section 3.3 Matrisome in Age-related Diseases. In this section the categories A, B and C do not appear to be clearly defined. What does “In category (B), we grouped common diseases that are more or less unlikely to be affected by ECM” mean? More or less is difficult to quantify or categorise and A, B and C appear to be very subjectively defined.

5.     Section 3.5. Potential Strategies Using Matrisome for Drug Repurposing. It is not clear how the authors propose that the drug-matrisome relationships would work. Please provide examples of how the relationships were identified and how they could be used to target the ECM SNVs mechanistically.

Other issues

1.     Materials and Methods section is very brief. Have you supplied sufficient information to permit others to perform equivalent analyses?

2.     Materials and Methods line 92. Do you provide a comprehensive list of the publicly-available datasets used?

3.     Typo line 113? 3’144. Be consistent with numerical formatting i.e. 3’144 or 3144.

4.     Supplementary tables and data would benefit from additional information (or formatting) in the provided excel files. For example, this reviewer always finds it useful to have a full legend with description of column headings in the appropriate excel file itself. It makes it much easier to interpret what is in each table.

5.     Typo line 125: adhesosome should be adhesome? You have not introduced this concept either.

6.     Line 127 the integrin subunit ITGB7 is not considered a collagen-binding integrin

7.     Line 185 the URL http://big.stats.ox.ac.uk/about is not active and results in an error. Please check for updated link URL

8.     Figures 5 and 7 are very hard to read in any detail as the text and details are too small to read. In their current format they are not useful and their content should be reconsidered.

9.     Lines 308-309. The terms “15 thousand” and “38 thousand” are mixed number formats. Write as text or as a number.

10.  Line 304 typo: “3.5. Potential Strategiues Using Matrisome for Drug Repurposing” should be “strategies”.

Author Response

Please see the attachment for response letter.

Reviewer 2 Report

Overall, the manuscript was well-written and scientifically sound. The authors do a great job establishing changes in the phenotypic and genotypic changes in aging and disease. This is a very relevant topic and deserves attention. Only minor criticisms can be made regarding grammar and syntax, which can be corrected with a thorough read thru.

Major concerns stemmed from the figures. They are simply illegible. Choice of font, color, size, etc. diminished the statement the authors were trying to make. Even when enlarged to 200% figures were unreadable. Please try to incorporate better quality images. Additionally, the overuse of references to supplementary data was extreme. If the authors continually refer to supplementary material, then that material may need to be incorporated into the manuscript. 

Author Response

(The authors gave the same response as above.)
